# Self-Corrected Multimodal Large Language Model for Robot Manipulation and Reflection

## Abstract

Multimodal Large Language Models (MLLMs) have demonstrated potential in visual instruction following across various tasks. Recently, some studies have integrated MLLMs into robotic manipulation, allowing robots to interpret multimodal information and predict low-level actions. While MLLM-based policies have shown promising progress, they may predict failure execution poses when faced with novel tasks or categories. To emulate human-like reasoning modes for more robust manipulation, we propose a Self-Corrected (SC)-MLLM. Our model combines fast system reasoning for directly predicting end-effector poses with slow system reasoning for reflecting on and correcting failure actions. For the fast system, we introduce parameter-efficient fine-tuning to empower MLLM with pose prediction capabilities, reframing this as a language modeling problem. For the slow system, when facing execution failures, our model learns to detect the causes of low-level action errors (i.e., position and rotation errors) and adaptively seeks prompt feedback from experts. Based on the feedback, SC-MLLM reflects on the current failure case and attempts to generate the corrected actions. Furthermore, we design a continuous policy learning method using successfully corrected samples, enhancing the model's adaptability to the current scene configuration and reducing the frequency of expert intervention. To evaluate our method, we conduct extensive experiments in both simulation and real-world settings. SC-MLLM significantly improves manipulation accuracy compared to previous state-of-the-art MLLM-based policy (ManipLLM), increasing from 57% to 79% on seen object categories and from 47% to 69% on unseen novel categories. Our project web page: https://sites.google.com/view/sc-mllm

## 1 Introduction

Recently, Multimodal Large Language Models (MLLMs) (Li et al., 2022; Alayrac et al., 2022; Liu et al., 2023b; Zhang et al., 2023; Gao et al., 2024; Lin et al., 2023; Zhang et al., 2024) have showcased remarkable abilities in visual instruction following and common sense reasoning. Some studies (Singh et al., 2023; Ahn et al., 2022; Liang et al., 2023; Driess et al., 2023) integrate MLLMs into robot manipulation, enabling robots to explore multimodal information and formulate task planning. Concurrently, other researchers (Brohan et al., 2022; Zitkovich et al., 2023; Li et al., 2023c;b; Kim et al., 2024; Liu et al., 2024b) are focusing on developing MLLMs capable of predicting robotic low-level action poses. While the integration of MLLMs into robotics has made encouraging strides, the current pipelines may lead to failure predictions when faced with novel tasks or object instances. Most prior research has overlooked the detection and correction of failure actions within the control process. This limitation significantly impacts their practicality in real-world settings, where uncertainties and unexpected obstacles are prevalent.

Recognizing the crucial role of self-correction in robot manipulation, recent studies have introduced some solutions. REFLECT (Liu et al., 2023c) stands out by utilizing LLMs to generate failure explanations and assist a language-based planner in correcting errors. Building on this innovation, subsequent research (Guo et al., 2023a; Skreta et al., 2023; Shi et al., 2024; Zha et al., 2023; Ming et al., 2023; Yu et al., 2023) explores robot correction through a hierarchical framework design. These approaches employ LLMs to correct high-level task planning while utilizing action models to execute low-level actions. However, the existing correction methods still face two main challenges. **1) Error Accumulation in the Hierarchical Framework.** Correction LLMs and the action model

Figure 1: Overview of our proposed SC-MLLM. **Fast system.** SC-MLLM reframes pose prediction as a language modeling problem, utilizing the initial state image and text prompts to generate the action pose. **Slow system.** When a failure occurs, SC-MLLM utilizes the end-state image and end-effector states for failure recognition, intelligently requesting prompt feedback from experts to reflect on and correct the action pose. **Continuous policy learning.** When an action is successfully corrected, SC-MLLM continuously learns policies from these samples, enhancing the manipulation accuracy of the fast system prediction.

are separate components, connected through prompts. For instance, although LLMs can generate correction prompts like "Move a little bit to the right," these prompts are often difficult for the action model to understand and translate into specific pose predictions. **2) Lack of Ability to Correct Low-Level SE(3) Poses.** Existing methods are unable to directly correct the end-effector poses of atomic tasks, which are fundamental for completing manipulation tasks. Additionally, they fail to learn from successful corrected cases and improve the policy model. Given these challenges, we raise a question: *"Can we develop an end-to-end robotic agent that not only possesses manipulation skills but also effectively corrects low-level failure actions?"*

Drawing inspiration from Daniel Kahneman's assertion that "human thinking is divided into a fast system and a slow system, which separately represent intuitive processes and more logical reasoning," (Kahneman, 2011) we introduce a self-corrected (SC)-MLLM that mimics a human-like thinking paradigm to address the above question. As shown in Figure 1, the SC-MLLM possesses both the fast system capability to directly predict end-effector poses and the slow system ability to reflect on and correct failure actions. For fast system construction, we transform low-level manipulation action into a language modeling problem, using language to directly generate SE(3) poses. During the training process, we parameter-efficient fine-tune the LLM to empower SC-MLLM with pose prediction capability. According to the RGB image and a text prompt, our SC-MLLM generates the end-effector position and rotation. To enhance the SC-MLLM with slow system capabilities, we propose a Chain-of-Thought training strategy for failure action reflection and correction based on expert feedback. Specifically, the SC-MLLM first recognizes SE(3) pose errors as position, rotation, or combined errors by utilizing the final state image and the robot's end-effector pose. Based on the identified error type, the SC-MLLM adaptively requests correction feedback from experts, such as position Mo et al. (2021), rotation Fang et al. (2023), and reasoning expert Achiam et al. (2023). Drawing on insights from previous errors and expert prompts, the SC-MLLM reflects on the current failure scenario and regenerates the appropriate contact pose.

During inference, the SC-MLLM initially interacts with the physical world using the fast system, activating the slow system for reflection and correction only when a failure action occurs. Moreover, as shown in Figure 1 c), we design a continuous policy learning method to transfer successfully corrected samples from the slow system to the fast system, utilizing exponential moving average techniques (Tarvainen & Valpola, 2017b) to enhance the pose prediction accuracy of the fast system. This method enhances the model's adaptability to the current scene configuration while also reducing the frequency of expert intervention.

To train our SC-MLLM, we generate 12k manipulation success samples, 15k failure samples, and corresponding 60k correction prompts in the SAPIEN simulation (Xiang et al., 2020). To systematically evaluate our method, we conduct extensive experiments in both simulation and real-

world datasets. In the SAPIEN simulator, SC-MLLM with slow system reasoning demonstrates an improvement in the manipulation success rate from 66% to 87% in seen categories and from 30% to 68% in unseen categories. After continual policy learning, SC-MLLM with fast system reasoning can also achieve 79% and 69% accuracy on seen and unseen categories, respectively. To validate the generalizability of our method, we further conduct closed-loop experiments in the RLBench simulation (James et al., 2020) and confirm its effectiveness through real-world experiments (as shown in the supplementary video). In summary, our contributions are as follows:

- To mimic a human-like thinking paradigm in manipulation, we propose a Self-Corrected (SC)-MLLM, equipping our model with not only the fast system ability to predict end-effector poses but also the slow system ability to reflect on and correct failure actions.

- For the slow system, we propose a Chain-of-Thought training strategy for detecting, reflecting on, and correcting low-level failure actions. SC-MLLM can adaptively request expert prompt feedback to regenerate the contact pose.

- During inference, we introduce a continuous policy learning method for successful corrected samples, continually enhancing the model's adaptability to scene configurations and reducing the frequency of expert intervention.

## 2 RELATED WORK

**Multimodal Large Language Models** Large language models (LLMs) have shown impressive reasoning capacities in a variety of downstream tasks (Touvron et al., 2023; Floridi & Chiriatti, 2020). In tackling complicated multi-modal reasoning problems, Multimodal Large language models (MLLMs) (Alayrac et al., 2022; Li et al., 2022; Zhang et al., 2023) have demonstrated impressive capacities in bridging modalities. BLIP (Li et al., 2022; 2023a) pre-trains encoder-decoder models using a dataset from image-text pairs, adding synthetic captions and filtering noisy ones for better vision-language understanding and generation. LLaVA (Liu et al., 2023a) and MiniGPT- 4 (Zhu et al., 2023) propose using a simple fully connected layer as a bridge between the image encoder and LLM. They also investigate the importance of dataset pre-training and instruction tuning. Meanwhile, some MLLMs (Chen et al., 2023; Wang et al., 2023c; Lin et al., 2023; Wang et al., 2023a) introduce vision-centric tasks on top of visual instruction tuning, validating that MLLMs are capable of producing fine-grained perceptual results. Besides, the introduction of 3D MLLMs (Guo et al., 2023b; Hong et al., 2023; Wang et al., 2023d; Yang et al., 2023b) aims to broaden the scope of reasoning and conversational capabilities obtained from LLMs to encompass the 3D modality.

**Robotic Manipulation.** Robotic manipulation has become a pivotal area of research due to its wide-ranging applicability. State-based reinforcement learning is a popular approach in this field (Joshi et al., 2020; Andrychowicz et al., 2020; Yarats et al., 2021; Geng et al., 2023b). While some studies have explored using the pure state as the policy input (Andrychowicz et al., 2020), more intricate scenarios require vision-based observation (Mo et al., 2019a; 2021; Liu et al., 2024a; Xu et al., 2022; Eisner et al., 2022; Wu et al., 2021; Huang et al., 2023; Zitkovich et al., 2023; Xu et al., 2023; Wan et al., 2023; Gong et al., 2023; Yang et al., 2023a; Wang et al., 2023b; Geng et al., 2023a) to perceive the environment and comprehend complex scenes and objects (Deng et al., 2023; Lei et al., 2023). Inspired by MLLMs success in general scenarios (Alayrac et al., 2022; Li et al., 2022; Zhang et al., 2023; Guo et al., 2023b; Li et al., 2024), some works have tried to employ the common sense reasoning ability of MLLMs to solve manipulation tasks. Palm-E (Driess et al., 2023) trains multimodal encodings end-to-end in conjunction with LLMs for manipulation planning. VoxPoser (Huang et al., 2023) extracts affordances and constraints from MLLMs to further zero-shot generate trajectories for manipulation tasks. RT-2 (Zitkovich et al., 2023), which transfers information to actions, holds promise for adapting more rapidly to novel situations. Robotflamingo (Li et al., 2023c) tries to fine-tune MLLM on imitation learning datasets to complete basic long-horizon tasks. Recent works (Li et al., 2023b; Liu et al., 2024b) further employ the reasoning ability of MLLMs and equip them with the ability to predict end-effector poses. Although integrating MLLMs into robotics has shown promising progress, current pipelines may lead to failure predictions when facing novel tasks or object instances.

**Robotic Failure Correction.** Several studies have delved into correcting robotic failures. RE-FLECT (Liu et al., 2023c) introduces LLMs for reasoning based on a summary of past experiences, utilizing failure explanations for improved planning. MULTIREACT (Yu et al., 2023) utilizes a

vision-language model (Radford et al., 2021) as a reward model to recognize and autonomously recover from intermediate execution failures. DoReMi (Guo et al., 2023a) conducts immediate detection of misalignments between plans and execution using LLMs and then recovers from them. CLAIRify (Skreta et al., 2023) generates iterative prompting with program verification to ensure action plans are valid. While these works demonstrate the use of LLMs in correcting execution failures in robotic tasks, they are limited to directly correcting low-level actions (e.g., 6 DoF pose) and fail to learn from the corrected feedback. In this paper, we aim to develop a slow system that mimics human-like thinking by enabling MLLMs to autonomously recognize and correct end-effector poses, while continually learning from the corrected samples.

## 3 METHOD

In Section 3.1, we introduce our problem formulation. Subsequently, in Section 3.2, we present our proposed Self-corrected (SC)-MLLM, detailing the model architecture design and the process of equipping our model with fast system and slow system abilities. Finally, we explain the details of the continuous policy learning mechanism in Section 3.3.

### 3.1 PROBLEM FORMULATION

In this paper, we make the first attempt to enhance the MLLM with a human-like thinking paradigm for robot manipulation, integrating both a fast system for predicting end-effector poses and a slower system for correcting failed actions. Therefore, our problem formulation comprises two components.

**End-effector pose prediction.** For manipulation ability, our SC-MLLM policy $\pi$ generates an action $a_i$ based on the image ($I_i \in \mathbb{R}^{W \times H \times 3}$) and language question ($L_i$) at the initial state. This action, denoted as $\pi(I_i, L_i) \to a_i$, corresponds to the 6-DoF control of a Franka Panda robot arm (Li et al., 2023b; Xu et al., 2022). It is parameterized by the end-effector position and rotation, where $a_i = (a_i^{\text{pos}}, a_i^{\text{rot}})$, with $a_i^{\text{pos}} \in \mathbb{R}^3$ representing a 3D coordinate and $a_i^{\text{rot}} \in \mathbb{R}^{3 \times 3}$ representing a rotation matrix. We include the end-effector pose in the language for MLLM fine-tuning.

**Failure reflection and correction.** During the actual manipulation process, robot action $a_i$ often encounters failures, known as error action $a_i^{err}$. For failure correction ability, our SC-MLLM policy $\pi$ utilize the end state RGB image ($I_e \in \mathbb{R}^{W \times H \times 3}$) and the language-descriptive robot state ($L_i^{a_i}$) to identify failure cases, represented as $\pi(I_e, L_i^{a_i}) \to c_i$. Depending on the error type $c_i$, our SC-MLLM can dynamically request prompt feedback $f_i$ from experts. This feedback is then utilized as input to re-predict actions $a_c$ using our SC-MLLM policy $\pi(I_i, f_i) \to a_c$. Details of the reflection and correction process are provided in Section 3.2.3.

### 3.2 SELF-CORRECTED MLLM

#### 3.2.1 MODEL ARCHITECTURE

To equip our model with foundational reasoning abilities, we adopt LLaMA-Adapter V2 (Gao et al., 2023) as our base MLLM. As shown in Figure 2, when presented with an RGB image, we utilize CLIP visual encoder (Radford et al., 2021) to extract visual features. Simultaneously, text prompts are encoded into textual features using the pre-trained LLaMA tokenizer (Touvron et al., 2023). Subsequently, our MLLM employs a projection layer to align visual tokens with LLM's token embedding, enabling the LLaMA to perform multimodal comprehension and generate corresponding answers. During the training process, we only fine-tune the injected adapters (Hu et al., 2021) within the LLM, while keeping the major pre-trained parameters frozen. This strategy is aimed at preserving the robust capabilities of the existing MLLM while enhancing the model with additional functionalities in manipulation and failure correction.

#### 3.2.2 FAST SYSTEM: END-EFFECTOR POSE PREDICTION

This part aims to enable our SC-MLLM with the fast system ability to directly generate end-effector poses. During the pre-collection of training data, we capture RGB images and their corresponding end-effector poses in the simulator (Xiang et al., 2020; James et al., 2020) when the manipulation is successful. During fine-tuning, as shown in the fast system part of Figure 2, we structure the

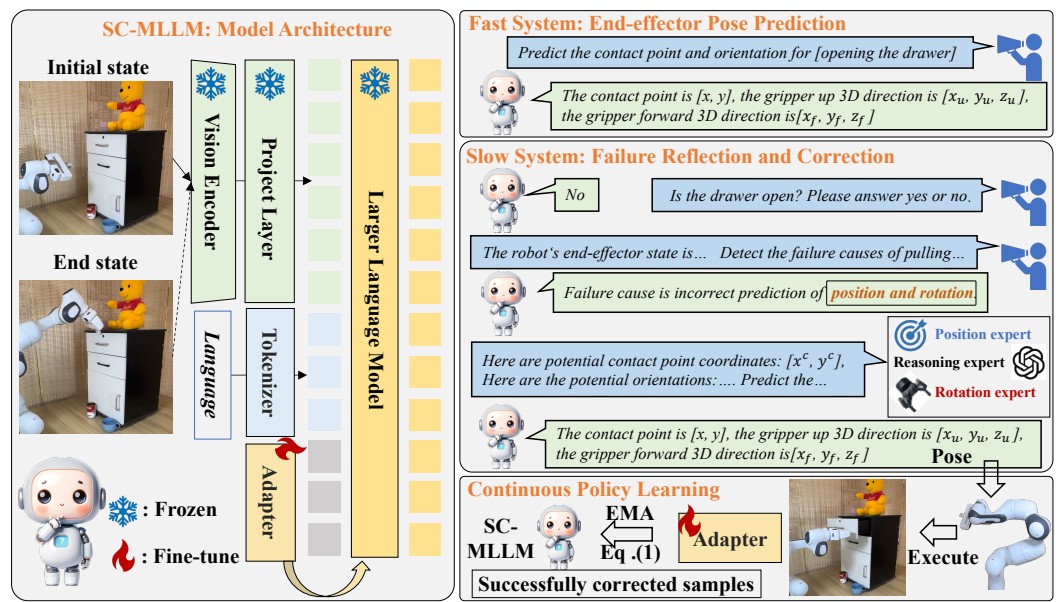

Figure 2: **Overall framework of SC-MLLM.** In the left part, we present the model architecture of SC-MLLM, which utilizes a vision encoder and projection layer to transform robot images into the LLM's language embeddings. These embeddings are then concatenated with text tokens and fed into the LLM, allowing it to directly generate end-effector poses through language responses. In the right section, we showcase the training strategies for SC-MLLM, which involve both the fast and slow system. SC-MLLM interacts with the object using the fast system to predict pose, while the slow system is activated only when a failure action occurs for reflection and correction. Notably, SC-MLLM employs the same prompt format during training and inference. Once the slow system successfully corrects failure samples, we introduce the continuous policy learning method to fine-tune the adapters in SC-MLLM, enhancing the model's manipulation accuracy under the fast system.

input text prompt for pose prediction as "*Predict the contact point and orientation for [task name].*" The answer format is specified as "*The contact point is [x, y], the gripper up 3D direction is [$x_u$, $y_u$, $z_u$], and the gripper forward 3D direction is [$x_f$, $y_f$, $z_f$].*" To simplify the direction regression prediction (Li et al., 2023b; Zitkovich et al., 2023), we convert it into a classification problem by discretizing the continuous numbers in the 100 normalized integer vectors into (-50, 50] discrete bins.

In the open-loop task, our SC-MLLM focuses on predicting the 2D coordinates $[x, y]$ of the contact pixel in the image, which are then translated into 3D space using depth information. We also derive the gripper's left direction (gripper z-forward) from its up and forward orientations based on geometric relationships. For the closed-loop task, instead of directly predicting the final contact pose of the end-effector, we continuously input the current state image to predict the trajectory. Additionally, we convert the robot's current state into language and combine it with the question input. The results of the open-loop and closed-loop experiments are presented in Section 4.2 and Appendix F, respectively.

### 3.2.3 SLOW SYSTEM: FAILURE REFLECTION AND CORRECTION

This section aims to equip our SC-MLLM with the slow system's ability to reflect on and correct failure actions. Different from previous correction works (Liu et al., 2023c; Ming et al., 2023) aimed at correcting high-level planning, we make the first attempt to directly correct the end-effector's 6-DoF control action through a chain of thought training strategy. For both open-loop and closed-loop tasks, we make pose corrections when the gripper interacts with the object. Before moving on to the correction process, it's necessary to assess whether the action has failed and identify the causes of the failure. Specifically, we leverage our model's inherent visual understanding capabilities, along with specific prompts, to determine whether the task has been completed. As shown in the slow system of Figure 2, we feed the end-state image into our SC-MLLM and pose the question, "*Is the drawer open? Please answer yes or no.*" If SC-MLLM determines that the current task is incomplete, it will subsequently reflect on the failure case. Since the pre-trained MLLM lacks failure recognition

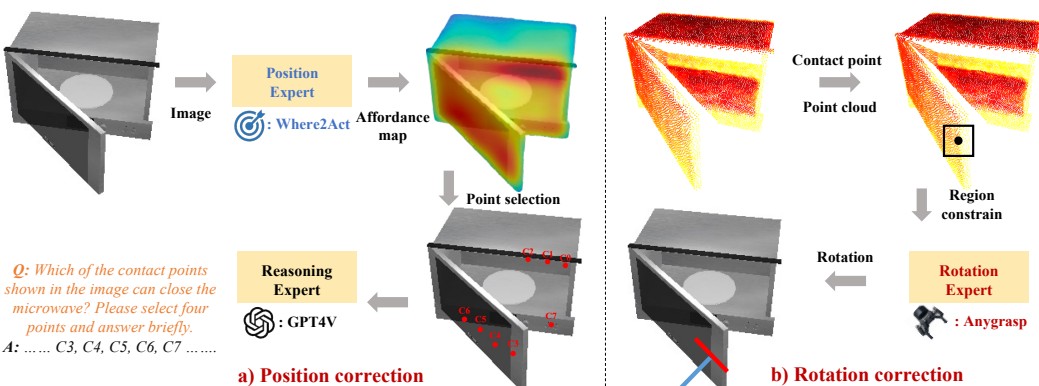

Figure 3: **The process of expert feedback.** For position correction, we input the end-state image into the position expert (Mo et al., 2021) to create an affordance map. Next, we randomly select points from areas with high affordance scores. These selected points are then projected back onto the image and fed into the reasoning expert (Achiam et al., 2023) for potential contact point selection. For rotation correction, we automatically select the manipulation area (bounding box) based on the contact point predicted by our model. Subsequently, we input the point cloud into the rotation expert (Fang et al., 2023) to predict potential action rotations within the selected manipulation areas.

capabilities, we collected 15,000 failure samples and 60,000 corresponding correction prompts in the simulator. Specifically, we categorize the failure cases into position, rotation, and combined errors. For position errors, we assess whether the contact point falls within the object's affordance region. For rotation errors, we evaluate whether the angle between the predicted Z axis and the normal of the object's movable plane exceeds 30 degrees. Both the affordance region and the object's movable parts are automatically generated from the simulator, allowing us to rapidly collect failure samples and classify the causes of errors. Then, we fine-tune the injected adapter of our model using the failure detection prompts shown in Figure 2. By inputting the end-state image and robot state, our model accurately identifies the causes of failure.

Based on the type of error, our SC-MLLM can adaptively request prompt feedback from specified experts and utilize this feedback to re-predict the action pose, as shown in Figure 2. To obtain position expert feedback during inference, as shown in Figure 3 a), we first use Where2Act (Mo et al., 2021) to generate an affordance map by inputting 2D images. The affordance map indicates the probability of achieving a moving distance when operating on certain pixels, reflecting the manipulable region of objects. However, existing affordance-based methods (Mo et al., 2019a; Eisner et al., 2022) can only predict a potential manipulable region with limited accuracy. For example, in Figure 3, the highest affordance score region predicted by the expert is C0-C3, which is obviously not a manipulable region. Therefore, to provide a more reliable contact point, we utilize the common sense reasoning ability of a reasoning expert (e.g., GPT-4V (Achiam et al., 2023)) to further filter the contact points. Specifically, we randomly select points from areas with high affordance scores. These selected points are then projected back onto the image and fed into the reasoning expert, using the following prompt to filter: "*Which of the contact points shown in the image can close the object? Please select n points* ..." This process allows us to automatically obtain relatively accurate contact points, which are then used as input prompts for our SC-MLLM.

As shown in Figure 3 b), we use Anygrasp (Fang et al., 2023) to generate the rotation correction prompt. Anygrasp is trained on large-scale data and directly inputs 3D point cloud data, which includes rich contextual information, enabling accurate grasp rotation prediction. However, our actions are not limited to grasping, so we cannot predict the position and rotation for the entire object. This is why we do not use Anygrasp as the position expert. To generate the rotation for the local region of each object, we select a manipulation box based on the contact point predicted by our model and generate the rotation within the selected box region. For the manipulation box, we expand 5 pixels outward from the contact point as the center. If the failure case involves combined errors, we sequentially apply position and rotation corrections. The rotation expert uses the highest-confidence contact point provided by the position expert to generate the corresponding rotation. Finally, we combine the correction feedback for both position and rotation as input prompts for our model. To enable our SC-MLLM to understand expert feedback, we introduced a similar question format during

training. In summary, as shown in Figure 2, we empower SC-MLLM with reasoning capabilities through a Chain-of-Thought (CoT) training strategy, enabling it to reflect on and correct failures. During inference, our model can significantly improve the action accuracy for failure samples based on the slow system, as demonstrated in Section 4.2 and Appendix F.

### 3.3 CONTINUOUS POLICY LEARNING

To equip our model with the fast system's ability for direct manipulation and the slow system's ability for failure correction, we integrate pose prediction, failure detection, and failure correction data for co-fine-tuning. Following previous works (Zhang et al., 2023; Li et al., 2023b), all outputs from the LLM are supervised using the cross-entropy loss $\mathcal{L}_{ce}$. During inference, since the relative position of the robot and the object changes during each manipulation process, it cannot reuse previous expert feedback. Therefore, after obtaining successfully corrected samples, we design a continuous policy learning method (shown in Figure 2). Note that, we utilize the prompt format used for training the fast system. This method aims to enhance the fast system's pose prediction accuracy without relying on the slow system's CoT reasoning, as expert interventions incur additional inference time. However, during fine-tuning with new successful samples, there is a risk of catastrophic forgetting, which can result in a loss of accuracy on previously trained samples. Therefore, we explore the use of exponential moving average (EMA) (Tarvainen & Valpola, 2017b) to continually learn from new data, formulated as:

$$\pi^t = \alpha \pi^{t-1} + (1 - \alpha)\pi^t \tag{1}$$

Where $t$ is the time step, $\pi$ represents our model. We set the updating weight $\alpha = 0.999$ based on (Tarvainen & Valpola, 2017a). We evaluate the effectiveness of the EMA scheme in action continual learning, as shown in Appendix D. Finally, our SC-MLLM can perform repeated continuous policy learning sessions for successfully corrected samples, continually transferring the slow system's knowledge to the fast system and improving the model's prediction accuracy. This process efficiently generates customized manipulation policies for the specific scenario, rather than relying on a shared, low-accuracy policy.

## 4 EXPERIMENT

In this section, we conduct extensive experiments in both simulation and real-world settings. First, we introduce the experimental setup in Sec .4.1, including data collection, implementation details, and evaluation metrics. In Sec .4.2, we compare our SC-MLLM with previous baselines on the simulation dataset. The ablation study is presented in Sec .4.3, demonstrating the effectiveness of each component. Finally, we present the qualitative analysis of real-world experiments in Sec .4.4.

### 4.1 EXPERIMENT SETTING

**Data Collection.** We use the simulator to collect training data under open-loop and closed-loop control. For the open-loop experiment, we follow the data collection process of previous works (Mo et al., 2021; Li et al., 2023b), adopting the SAPIEN engine (Xiang et al., 2020) to set up an interactive simulation environment with articulated objects from PartNet-Mobility (Mo et al., 2019b). The Franka Panda Robot, equipped with a suction gripper, serves as the robotic actuator. During data collection, we randomly select a contact point **p** on the movable part and orient the end-effector's z-axis opposite to its normal vector, with a random y-axis direction to interact with the object. Successful operations are categorized as successful samples and integrated into the dataset. Our training dataset comprises 12k successful manipulation samples across 20 categories. Meanwhile, we collect 15k failure samples and 60k corresponding correction prompts, covering position, rotation, and combined errors. For evaluation, we generate 1k examples for the test set, comprising 20 training (seen) and 10 testing (unseen) categories. The unseen categories are used to evaluate the generalization capability of SC-MLLM. The additional description of the dataset and variation can be found in Appendix B.1 and B.2. For the closed-loop experiment, we select five tasks from RLBench (James et al., 2020), including "take USB out of computer," "close fridge," "close box," "toilet seat down," and "unplug charger." We utilize a single-view input (the front view from a third-person perspective) and follow to the key frame selection manner outlined in previous work Shridhar et al. (2023).

Table 1: Comparisons of our SC-MLLM against baseline methods. The table shows the performance of different methods across various seen and unseen categories. "Experts" refers to using our position and rotation experts to generate action poses in a zero-shot manner. "Fast" and "Slow" represent our method's results for the fast system's direct pose prediction and the slow system's corrected prediction based on expert prompts, respectively. "CPL" refers to the continuous policy learning method used to update our model. The representation for each task icon is shown in Table 2.

| Method | Seen Categories | | | | | | | | | | | | | | | |
|---|---|---|---|---|---|---|---|---|---|---|---|---|---|---|---|---|
| UMPNet Xu et al. (2022) | 0.23 | 0.36 | 0.41 | 0.22 | 0.24 | 0.30 | 0.43 | 0.34 | **0.51** | 0.21 | 0.66 | 0.27 | 0.23 | 0.23 | 0.29 | 0.60 |
| FlowBot3D Eisner et al. (2022) | 0.45 | 0.48 | 0.45 | 0.32 | 0.32 | 0.37 | 0.43 | 0.23 | 0.26 | 0.14 | 0.39 | 0.31 | 0.38 | 0.32 | 0.23 | 0.43 |
| ManipLLM Li et al. (2023b) | 0.72 | 0.56 | 0.32 | 0.79 | 0.48 | 0.53 | 0.66 | 0.69 | 0.39 | 0.52 | 0.53 | 0.4 | **0.64** | 0.73 | **0.62** | 0.52 |
| Experts | 0.34 | 0.36 | 0.33 | 0.44 | 0.45 | **0.56** | 0.32 | 0.19 | 0.48 | 0.28 | 0.53 | 0.29 | 0.27 | 0.32 | 0.27 | 0.45 |
| Ours(Fast) | 0.78 | 0.63 | 0.58 | 0.70 | 0.52 | 0.13 | 0.81 | 0.88 | 0.56 | 0.71 | 0.84 | 0.80 | 0.46 | 0.76 | 0.30 | 0.83 |
| Ours(Fast+Slow) | 0.97 | 0.90 | 0.66 | 0.93 | 0.95 | 0.66 | 0.97 | 0.96 | 0.87 | 0.92 | 0.90 | 0.87 | 0.78 | 0.94 | 0.30 | 0.90 |
| Ours(CPL+Fast) | **0.90** | **0.75** | 0.58 | **0.87** | **0.95** | 0.46 | **0.89** | **0.92** | 0.50 | **0.78** | **0.90** | **0.85** | 0.63 | **0.90** | 0.38 | **0.90** |

| Method | Seen Categories | | | | | Unseen Categories | | | | | | | | | | |
|---|---|---|---|---|---|---|---|---|---|---|---|---|---|---|---|---|
| | | | | | AVG | | | | | | | | | | | AVG |
| UMPNet Xu et al. (2022) | 0.32 | 0.30 | 0.11 | 0.58 | 0.34 | **0.36** | 0.36 | 0.38 | 0.47 | 0.21 | 0.12 | 0.24 | 0.23 | 0.28 | 0.12 | 0.28 |
| FlowBot3D Eisner et al. (2022) | 0.19 | 0.33 | 0.23 | 0.47 | 0.33 | 0.29 | 0.47 | 0.64 | 0.31 | 0.27 | 0.30 | 0.09 | 0.41 | 0.35 | 0.37 | 0.35 |
| ManipLLM Li et al. (2023b) | 0.39 | **0.75** | 0.44 | 0.67 | 0.57 | 0.32 | 0.22 | 0.65 | **0.69** | 0.38 | 0.85 | 0.27 | 0.53 | 0.26 | 0.38 | 0.47 |
| Experts | 0.21 | 0.49 | 0.29 | 0.24 | 0.36 | 0.33 | 0.36 | 0.49 | 0.36 | 0.19 | 0.42 | 0.28 | 0.41 | 0.47 | 0.56 | 0.39 |
| Ours(Fast) | 0.20 | 0.68 | 0.48 | 0.60 | 0.66 | 0.09 | 0.25 | 0.39 | 0.66 | 0.64 | 0.23 | 0.21 | 0.56 | 0.10 | 0.50 | 0.30 |
| Ours(Fast+Slow) | 0.80 | 0.89 | 0.74 | 0.98 | 0.87 | 0.27 | 0.65 | 0.71 | 0.83 | 0.85 | 0.71 | 0.73 | 0.87 | 0.48 | 0.90 | 0.68 |
| Ours(CPL+Fast) | **0.60** | 0.71 | **0.74** | **0.90** | **0.79** | 0.27 | **0.61** | **0.71** | 0.50 | **0.92** | 0.69 | 0.69 | 0.80 | 0.70 | 0.81 | 0.69 |

**Implementation Details.** Our SC-MLLM loads the pre-trained parameters of LLaMA-Adapter-v2 (Zhang et al., 2023), which includes a pre-trained CLIP (Radford et al., 2021) as the visual encoder, a 7B LLaMA (Touvron et al., 2023) model as the language model, and a multi-modal projection module consisting of 32 transformer layers. Throughout the fine-tuning phase, we utilize the Adam optimizer with $(\beta_1, \beta_2) = (0.9, 0.999)$ and an initial learning rate of 1e-4, with a warm-up period of one epoch. We fine-tuned our model on four 80G A100 GPUs for 10 epochs.

**Evaluation Metric.** For the open-loop experiment, we follow the metrics from previous works (Li et al., 2023b), using the manipulation success rate. Specifically, the object starts in its initial state, and the goal is to actuate the joint to its target state. We use the predicted contact point and rotation to interact with objects and complete the task. Similar to (Mo et al., 2021), the trajectory for each task is predefined, i.e., moving backward along the z-axis of the end-effector pose. An action is considered successful if the joint state difference before and after interaction exceeds a threshold of 0.1 meters. For the closed-loop experiment, we follow the evaluation metrics from previous works(James et al., 2020), assessing task success rate based on predefined success conditions.

## 4.2 MAIN RESULTS ON MANIPULATION

For the open-loop experiment, we compare our SC-MLLM against four representative baselines: UMPNet (Xu et al., 2022), Flowbot3D (Eisner et al., 2022), ManipLLM (Li et al., 2023b), and our utilized experts. Following previous work (Li et al., 2023b), we conduct comparisons based on the interaction pose. To ensure a fair comparison, all methods use the same train/test split and end-effector settings. Reproduction details of the baselines are provided in Appendix B.3. As shown in Table 1, using our combined fast and slow systems, Ours(Fast+Slow) achieves an impressive 87% accuracy on seen categories and 68% accuracy on unseen categories, significantly outperforming the other methods. Specifically, compared to the previous SOTA ManipLLM, Ours(Fast+Slow) achieves improvements of 30% and 21% in accuracy for seen and unseen categories, respectively. These results demonstrate that SC-MLLM can effectively correct failed actions through the reasoning capabilities of the slow system's chain of thought. Additionally, we compare SC-MLLM with our employed experts (zero-shot) to verify that the improvements are not solely due to the expert prompts but also result from our model's reflection and correction abilities. When comparing Ours(CPL+Fast) and Ours(Fast), we find that Ours(CPL+Fast) achieved improvements of 13% and 39% in accuracy for seen and unseen categories, respectively. This indicates that our method effectively enhances

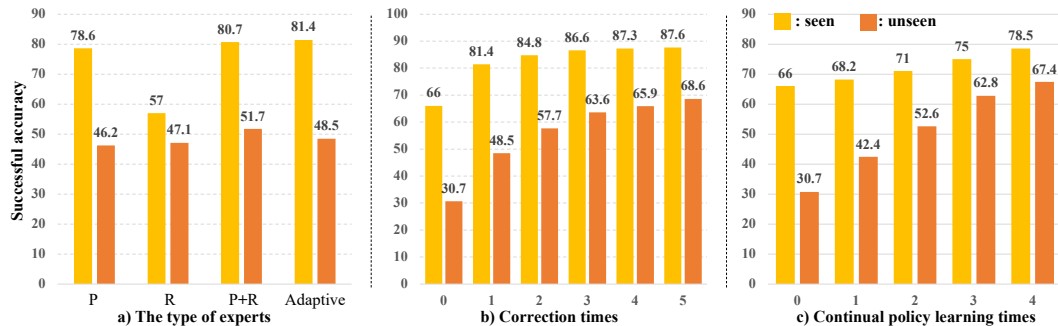

Figure 4: Ablation study for each method in our SC-MLLM framework. Part a) examines the impact of various expert feedback prompts on slow system reasoning. Part b) investigates the effects of different correction times on pose correction accuracy. Part c) analyzes the influence of continuous policy learning iterations on the model's prediction accuracy.

the model's manipulation accuracy and adaptability to varying scene configurations. The significant improvement observed in unseen categories further demonstrates that our proposed method can effectively enhance the model's generalization capabilities. In the closed-loop experiment, Ours(Fast) achieves an average success rate of 63% on the tasks "take USB out of computer," "close fridge," "close box," "toilet seat down," and "unplug charger." In contrast, Ours(Fast+Slow) achieves a success rate of 91%, validating the generalizability of our method in improving manipulation stability. Due to space limitations, we present the detailed closed-loop experiment table in Appendix F.

## 4.3 ABLATION STUDY

To elucidate the contribution and effectiveness of individual methods within our SC-MLLM, we conduct extensive ablation studies in the Sapien simulator.

**The impact of expert type.** First, we investigate the impact of various expert feedback prompts on the slow system's correction success rate. As shown in Figure 4 a), "P" and "R" represent utilizing our designed position and rotation feedback, respectively. "P + R" represents the utilization of combined expert feedback, while "Adaptive" refers to adaptively seek expert feedback based on the cause of failure. For all experiments, we input the prompt feedback into our SC-MLLM, allowing it to re-predict poses based on the prompt. We find that any type of expert prompt can improve unseen manipulation accuracy, demonstrating the importance of correction for novel object manipulation. Additionally, we observe that "Ours" achieves comparable results to "P + R", but with lower expert intervention costs. The results confirm that detecting failure cases and adaptively seeking expert feedback is essential for stable manipulation. The accuracy of failure detection is shown in Appendix C.

**The impact of correction times.** In the slow system, our utilized experts can return multiple prompts simultaneously; for example, the position expert can provide n potential contact points. Therefore, we explore the impact of different correction times on pose correction accuracy. As shown in Figure 4 b), the x-axis represents the number of expert prompts used for correction. Note that we count a manipulation as successful if it succeeds even once during multiple corrected actions. We find that with just one correction, SC-MLLM achieves an improvement of 15.4% on seen objects and 17.8% on unseen objects compared to direct prediction without expert feedback. The results demonstrate the effectiveness of our designed correction paradigm. Furthermore, once the number of corrections reaches three or more, the slow system of SC-MLLM achieves stable manipulation accuracy.

**The impact of continuous policy learning times** When the slow system corrects failure actions and re-predicts poses to accomplish the current task, we can employ the continuous policy learning method to update the model. The varying update iterations will impact the fine-tuning time and the manipulation accuracy achieved after fine-tuning. As shown in Figure 4 c), all manipulation results are evaluated using the fast system's inference manner after the continual learning process. The x-axis represents the number of continuous policy learning iterations. We find that both seen and unseen categories show significant improvement after multiple rounds of fine-tuning. The results confirm the effectiveness of our continuous learning method, efficiently transferring the slow system's knowledge to the fast system and improving the model's prediction accuracy. Finally, to demonstrate that the

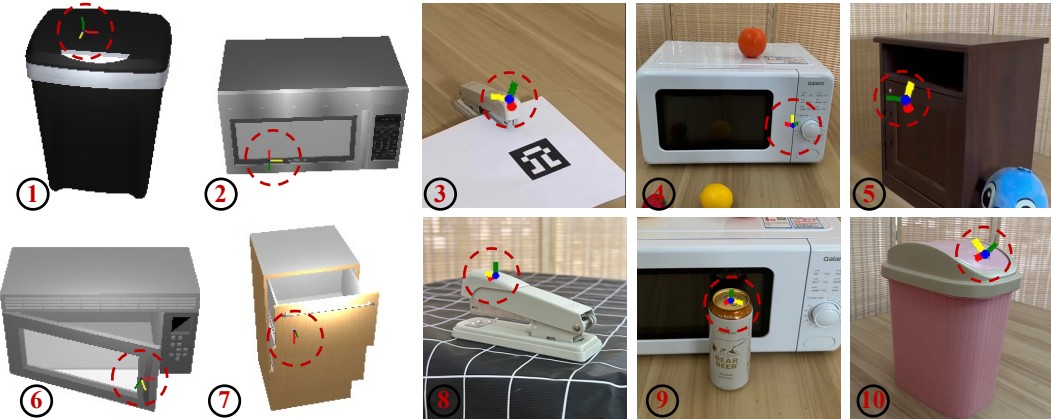

Figure 5: Visualization of pose predictions. The green, red, and yellow lines represent the z-axis, y-axis, and x-axis of the end-effector direction, while the blue dot indicates the contact point.

improvements are not due to overfitting on the test set, we also evaluate our model on the test B set after the continuous learning process (Appendix E). In test B, only the relative positions between the robot and the object are altered compared to the test set.

### 4.4 REAL-WORLD EVALUATION

We conduct real-world experiments involving interactions with various household objects using a Franka Emika robotic arm. For the open-loop experiment, we maintain consistency with the Sapien simulator by using a suction gripper as the actuator. For the closed-loop experiment, we align with RLBench by using a finger gripper as the actuator. Demonstration videos are shown in the supplementary material. Although ManipLLM (Li et al., 2023b) demonstrate that MLLM-based methods are less affected by the sim-to-real domain gap, we still made efforts to increase the diversity of simulation data collection. Specifically, we increase scenario diversity by varying elements such as object part poses, camera angles, lighting, and more to mitigate the potential sim-to-real gap. As shown in Figure 5, the left half presents the visualization results from the Sapien simulator, while the right half displays the visualization results from the real world. We project our corrected pose predictions back into the 2D image using camera parameters to indicate the pose that will contact the object. The green, red, and yellow lines represent the z-axis, y-axis, and x-axis of the end-effector direction, respectively, while the blue dot indicates the contact point. By employing the combined inference approach (fast system and slow system), SC-MLLM can accurately predict contact points and 3D directions in real-world scenarios. However, there are still some failure cases. For instance, in case number 2, shown in Figure 5, the predicted contact position is too close to the pivot of the microwave door. Similarly, in case number 9, the predicted pose is influenced by other objects in the image, which leads to task failure.

## 5 CONCLUSION AND LIMITATION

Drawing on Daniel Kahneman's concept that "human thinking is divided into a fast system and a slow system," we propose a self-correcting (SC) MLLM that emulates a human-like thinking model to achieve stable manipulation. The SC-MLLM integrates a fast system capable of directly predicting end-effector poses and a slow system designed to reflect on and correct failed actions. During inference, the fast system initially interacts with the physical world, while the slow system is only activated when a failure occurs, allowing for reflection and correction. Additionally, we introduce a continuous policy learning method to enable the SC-MLLM to learn from successfully corrected actions, continually improving the fast system's manipulation accuracy. As for limitations, the SC-MLLM's backbone relies on MLLM to leverage its chain-of-thought(CoT) reasoning capabilities. However, the MLLM leads to relatively slow control frequency, especially in closed-loop tasks. In future work, we plan to incorporate more efficient MLLMs (Wen et al., 2024; Liu et al., 2024b; Zhou et al., 2024), speeding up the CoT reasoning process of our slow system.

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

## A APPENDIX

Due to space limitations, we provide additional details on the dataset, baseline reproduction, SC-MLLM failure detection evaluation, and supplementary experiments. Below, we outline the contents of our appendix.

★ **Dataset & Baseline Reproduction Details**

- Dataset details & Dataset variation & Baseline reproduction details.

★ **Failure Detection Results**

- Experimental results and analysis of Failure Detection.

★ **Additional Ablation Studies**

- Experimental results of different continual learning methods.

★ **Experimental Results on Test B set**

- Experimental results of our SC-MLLM on Test B set.

★ **Details of Close-loop Experiments**

## B DATASET & BASELINE REPRODUCTION DETAILS

### B.1 SAPIEN DATASET DETAILS

As shown in Table 2, our training dataset comprises 12k manipulation scenarios, encompassing 20 distinct object categories, specifically including: Safe (*S*), Door (*D*), Display (*DS*), Refrigerator (*RF*), Laptop (*LT*), Lighter (*LI*), Microwave (*MW*), Mouse (*MO*), Box (*BX*), Trash Can (*TC*), Kitchen Pot (*KP*), Suitcase (*SU*), Pliers (*PL*), Storage Furniture (*SF*), Remote (*RM*), Bottle (*B*), Folding Chair (*FD*), Toaster (*TS*), Lamp (*L*), and Dispenser (*DP*). Following Partnet (Mo et al., 2019b), different tasks are designed for each category. For instance, opening the door or control panel of a refrigerator, opening the cap of a bottle, and rotating the lid of a box. The detailed task design can be found at: https://sapien.ucsd.edu/browse

In performance evaluation experiments, we utilize two primary test datasets: Test set and Test B set. Both datasets consist of 1081 manipulation scenarios and include 30 object categories, as detailed in Table 2. We collect the test B set where only the relative positions between the robot and the object are altered compared to the test set. This test B set aims to validate the effectiveness of our continuous policy learning, ensuring it does not overfit on the test set. Among these, 20 categories are present in the training set (seen), while 10 categories are not included in the training set (unseen), which are: Toilet (*TL*), Scissors (*SC*), Table (*T*), Stapler (*ST*), Kettle (*K*), USB (*U*), Switch (*SW*), Washing Machine (*WM*), Faucet (*FC*), and Phone (*PH*). This setup aims to thoroughly assess the model's generalization capabilities.

Table 2: Representation of each category icon.

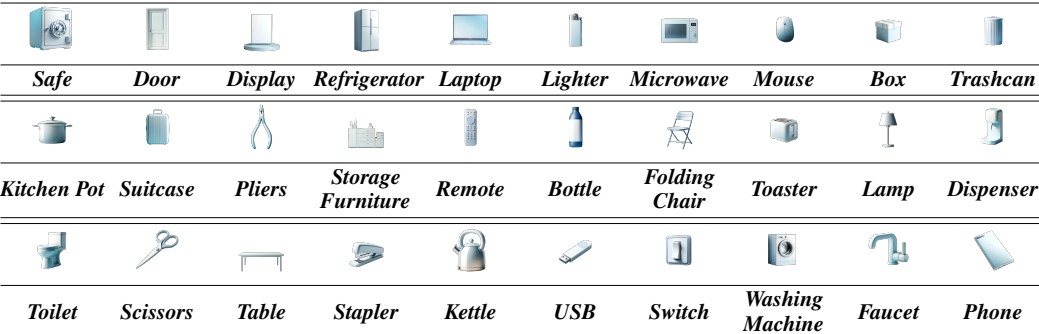

| Safe | Door | Display | Refrigerator | Laptop | Lighter | Microwave | Mouse | Box | Trashcan |
|---|---|---|---|---|---|---|---|---|---|
| Kitchen Pot | Suitcase | Pliers | Storage Furniture | Remote | Bottle | Folding Chair | Toaster | Lamp | Dispenser |
| Toilet | Scissors | Table | Stapler | Kettle | USB | Switch | Washing Machine | Faucet | Phone |

## B.2 SAPIEN DATASET VARIATION

Regarding the variation between training and testing data, we followed the data collection settings of where2act (Mo et al., 2021) and ManipLLM (Li et al., 2023b). The specific variations can be divided into two aspects: 1) Asset Variation and 2) State Variation.

1) Asset Variation: We use 20 categories from PartNet (Mo et al., 2019b) for seen objects and reserve the remaining 10 categories for unseen objects to analyze if RoboMamba can generalize to novel categories. Specifically, we further divide the seen objects into 1037 training shapes and 489 testing shapes, using only the training shapes to construct the training data. Thus, the shapes of the seen objects encountered during training and testing are different. For unseen categories, there are a total of 274 shapes, which are used exclusively in the testing data.

2) State Variation: We observe the object in the scene from an RGB-D camera with known intrinsics, mounted 4.5-5.5 units away from the object, facing its center. The camera is located at the upper hemisphere of the object with a random azimuth between [-45, 45] and a random altitude between [30, 60]. Since the tasks involve 'pulling,' we also initialize the starting pose for each articulated part randomly between its rest joint state (fully closed) and any position up to half of its joint state (half-opened). These state settings are utilized for both training and testing data, aiming to boost the model's generalization ability.

## B.3 BASELINE REPRODUCTION

**UMPNet (Xu et al., 2022)**: It inputs visual observations, such as RGB images and depth maps, of an articulated object to generate a sequence of actions in SE(3) space. It identifies the correct position on the object to interact with (*e.g.*, interacting with the lid rather than the base) and determines the appropriate action direction (*e.g.*, pulling up instead of pushing down) for interaction.

**Flowbot3D (Eisner et al., 2022)**: Flowbot3D begins by observing the initial configuration of the object of interest in the form of point cloud data. It is then post-processed and inputted into the model, which predicts 3D flow vectors for each point. It selects the point with the maximum flow vector magnitude and uses motion planning to make contact with that point via suction based on the selected flow.

**ManipLLM (Li et al., 2023b)**: ManipLLM uses chain-of-thought reasoning to enable the model to precisely generate an initial contact end-effector pose, including the pixel coordinates, gripper upward direction, and gripper forward direction. It then employs an active impedance adaptation policy that adjusts the direction based on force feedback to ensure a smooth movement trajectory.

## C FAILURE DETECTION

In this paper, we introduce a self-corrected (SC) MLLM that mimics a human-like thinking paradigm, including two reasoning modes: fast system and slow system. SC-MLLM possesses both the fast system's capability to directly predict end-effector poses and the slow system's ability to reflect on and correct failure actions. In the slow system chain of thought reasoning, one important intermediate process is failure case detection. The failure detection results are shown in Table 3. The experiments are evaluated on a self-collected dataset which contains 1K failure test samples on 20 seen categories of objects with 3 failure causes (i.e., position, rotation, or combined error). And

Table 3: Failure detection accuracy is evaluate on collected failure manipulation samples across 20 seen categories with three failure causes: position, rotation, or both.

| Failure Causes | Object Categories | | | | | | | | | | | | | | | | | | | | AVG |
|---|---|---|---|---|---|---|---|---|---|---|---|---|---|---|---|---|---|---|---|---|---|
| *Rotation* | 0.95 | 0.84 | 0.88 | 0.80 | 0.81 | 0.96 | 0.88 | 0.82 | 0.94 | 0.78 | 0.93 | 0.73 | 0.83 | 0.89 | 0.93 | 0.94 | 1.00 | 0.82 | 0.95 | 0.95 | **0.89** |
| *Position* | 0.95 | 1.00 | 1.00 | 1.00 | 1.00 | 1.00 | 0.95 | 1.00 | 0.94 | 1.00 | 0.95 | 1.00 | 1.00 | 0.93 | 1.00 | 0.91 | 1.00 | 1.00 | 1.00 | 0.95 | **0.98** |
| *Position & Rotation* | 0.71 | 0.74 | 0.92 | 0.88 | 0.83 | 1.00 | 0.71 | 0.91 | 0.91 | 0.95 | 0.83 | 0.82 | 0.82 | 0.82 | 0.92 | 0.84 | 1.00 | 0.84 | 1.00 | 0.61 | **0.84** |

SC-MLLM achieves an impressive average accuracy of **0.89** for rotation-related failures. This high accuracy underscores the model's capability to precisely identify and diagnose issues related to

the robot's manipulation direction. Notably, categories such as *Remote*, *Lamp*, and *Laptop* exhibit exceptional accuracy rates, reflecting the model's robustness across varied object geometries and complexities.

For position-related failures, SC-MLLM achieves an outstanding average accuracy of **0.98**. This near-perfect detection rate emphasizes the reliability of our SC-MLLM in pinpointing positional inaccuracies during manipulation actions. Categories including *Mouse*, *Phone* , *Pliers*, and *Refrigerator* all attain a flawless accuracy of 1.00, demonstrating the SC-MLLM's precision in handling positional errors.

When addressing failures caused by both position and rotation, the model maintains a commendable average accuracy of **0.84**. Despite the increased complexity of these cases, our method effectively diagnoses combined failures, ensuring that subsequent correction experts are accurately requested. Categories such as *Remote*, *Kettle*, and *Laptop* once again exhibit high accuracy rates, reaffirming the model's adaptability and precision.

These results highlight the effectiveness of our SC-MLLM in failure case detection and reflection. Accurately identifying the failure cause can significantly enhance the stability of our slow system reasoning, thereby improving failure correction.

## D  ADDITIONAL ABLATION STUDY

To further validate the robustness of our proposed SC-MLLM, we conduct supplementary ablation studies. In this section, we assess the efficacy of our approach in the context of continual policy learning. Specifically, we integrate various continual learning methods into our self-correction process to update the model and evaluate the performance of the updated model. The test set used for this evaluation, is consistent with the experimental setup described in the main text. It comprises 30 categories of manipulation targets, totaling 1081 manipulation scenarios. The results of these evaluations are presented in Table 4, demonstrating the performance improvements achieved through the integration of continual learning methods in our method. After each update, the model undergoes a new round of manipulation tests, and successful manipulation instances are collected. These successful examples were then incorporated as new data for continual learning, further updating the model. All test results are obtained after the base model is updated four times continually, with performance measured and recorded accordingly.

Table 4: Comparisons of our proposed method against other continual learning methods. The table presents the performance of different methods across various seen and unseen categories. "Ours(CPL+Fast)" represents our method's results for continually learned policies without slow system reasoning.

| Method | Seen Categories | | | | | | | | | | | | | | | |
|---|---|---|---|---|---|---|---|---|---|---|---|---|---|---|---|---|
| EWC Kirkpatrick et al. (2017) | 0.71 | 0.80 | 0.33 | 0.72 | 0.71 | 0.40 | 0.78 | 0.96 | 0.56 | 0.71 | 0.85 | 0.80 | 0.38 | 0.76 | 0.15 | 0.90 |
| LwF Li & Hoiem (2017) | 0.88 | 0.73 | 0.41 | 0.81 | 0.90 | 0.40 | 0.83 | 0.88 | 0.37 | 0.82 | 0.90 | 0.84 | 0.51 | 0.88 | 0.30 | 0.93 |
| Experience Replay Rolnick et al. (2019) | 0.83 | 0.73 | 0.50 | 0.84 | 0.85 | 0.26 | 0.72 | 0.88 | 0.56 | 0.75 | 0.90 | 0.79 | 0.42 | 0.83 | 0.23 | 0.90 |
| Ours(CPL+Fast) | 0.90 | 0.75 | 0.58 | 0.87 | 0.95 | 0.46 | 0.89 | 0.92 | 0.50 | 0.78 | 0.90 | 0.85 | 0.63 | 0.90 | 0.38 | 0.90 |

| Method | Seen Categories | | | | AVG | Unseen Categories | | | | | | | | | AVG |
|---|---|---|---|---|---|---|---|---|---|---|---|---|---|---|---|
| EWC Kirkpatrick et al. (2017) | 0.20 | 0.68 | 0.53 | 0.90 | 0.68 | 0.00 | 0.29 | 0.50 | 0.33 | 0.35 | 0.52 | 0.47 | 0.56 | 0.25 | 0.87 | 0.42 |
| LwF Li & Hoiem (2017) | 0.40 | 0.73 | 0.67 | 0.80 | 0.75 | 0.27 | 0.53 | 0.67 | 0.83 | 0.92 | 0.64 | 0.65 | 0.78 | 0.47 | 0.93 | 0.62 |
| Experience Replay Rolnick et al. (2019) | 0.00 | 0.70 | 0.61 | 0.80 | 0.71 | 0.00 | 0.40 | 0.61 | 0.66 | 0.57 | 0.58 | 0.56 | 0.68 | 0.34 | 0.93 | 0.52 |
| Ours(CPL+Fast) | 0.60 | 0.71 | 0.74 | 0.90 | 0.79 | 0.27 | 0.61 | 0.71 | 0.50 | 0.92 | 0.69 | 0.69 | 0.80 | 0.70 | 0.81 | 0.69 |

As shown in Table 4, Ours(CPL+Fast) seamlessly integrates Exponential Moving Average (EMA) to continually fine-tune the injected adapters, showing significant performance improvements across multiple evaluation metrics. Our method surpasses Elastic Weight Consolidation (EWC), Learning without Forgetting (LwF), and Experience Replay in both seen and unseen categories. Specifically, in the seen categories, our method achieves an impressive average successful rate of 0.79, compared to 0.68 for EWC, 0.75 for LwF, and 0.71 for Experience Replay. This superior performance indicates

that our approach not only retains knowledge from previous tasks but also effectively assimilates new information, thereby addressing the prevalent problem of catastrophic forgetting.

In the unseen categories, our method consistently maintains a high performance with an average score of 0.69, compared to 0.42 for EWC, 0.62 for LwF, and 0.52 for Experience Replay. This consistent performance across both seen and unseen categories underscores the robustness of our continual learning strategy. Notably, our method excels in categories such as *Laptop (LT)*, *Phone (PH)*, and *Refrigerator (RF)*, demonstrating its effectiveness to generalize across a diverse array of objects and scenarios.

Furthermore, our approach exhibits marked improvement in categories with lower baseline performances. For instance, in the *Dispenser (DP)* category, our method achieves a perfect score of 0.95, highlighting its capacity to manage challenging tasks with remarkable efficacy. Similarly, in the *Lamp (L)* and *Kettle (K)* categories, our method significantly outperforms other continual learning methods, achieving scores of 0.74 and 0.92, respectively. Overall, the experimental results validate the effectiveness of our proposed continuous policy learning. By incorporating the Exponential Moving Average scheme in the fine-tuning process, our approach ensures robust performance improvements while mitigating the risks of catastrophic forgetting. This enables our model to efficiently transfer successfully corrected samples from the slow system to the fast system.

# E  EXPERIMENTAL RESULTS ON TEST B SET

In this section, we evaluate our proposed method on the Test B set following the continuous learning process, aiming to demonstrate that the observed improvements are not simply due to overfitting on the test set. Similar to the Test set, the Test B set consists of 1,081 manipulation scenarios involving 30 different objects. To simulate real-world applications, we adjusted the relative positions between the robot and the objects in the Test B set compared to the Test set. It is important to note that the Test B set is not used for any fine-tuning. The model's performance on the Test B set is recorded after each iteration of continual policy learning on the Test set.

Table 5: The model's performance is evaluated on the Test B set. The success rate is recorded after each iteration of continual policy learning on the Test set.

| Method | Seen Categories | | | | | | | | | | | | | | | |
|---|---|---|---|---|---|---|---|---|---|---|---|---|---|---|---|---|
| Ours-cl-turn1 | 0.69 | 0.70 | 0.25 | 0.81 | 0.71 | 0.33 | 0.83 | 0.88 | 0.62 | 0.60 | 1.00 | 0.74 | 0.31 | 0.71 | 0.07 | 0.93 |
| Ours-cl-turn2 | 0.71 | 0.58 | 0.16 | 0.67 | 0.85 | 0.40 | 0.78 | 0.80 | 0.56 | 0.75 | 0.66 | 0.76 | 0.42 | 0.74 | 0.07 | 0.93 |
| Ours-cl-turn3 | 0.80 | 0.63 | 0.33 | 0.74 | 0.90 | 0.53 | 0.75 | 0.73 | 0.62 | 0.67 | 1.00 | 0.74 | 0.38 | 0.81 | 0.07 | 1.00 |
| Ours-cl-turn4 | 0.80 | 0.75 | 0.41 | 0.74 | 0.85 | 0.40 | 0.70 | 0.76 | 0.68 | 0.67 | 1.00 | 0.79 | 0.27 | 0.85 | 0.15 | 1.00 |

| Method | Seen Categories | | | | | Unseen Categories | | | | | | | | | |
|---|---|---|---|---|---|---|---|---|---|---|---|---|---|---|---|
| | | | | | AVG | | | | | | | | | | AVG |
| Ours-cl-turn1 | 0.00 | 0.56 | 0.29 | 0.60 | 0.61 | 0.00 | 0.26 | 0.35 | 0.00 | 0.71 | 0.26 | 0.65 | 0.53 | 0.12 | 0.62 | 0.32 |
| Ours-cl-turn2 | 0.00 | 0.56 | 0.33 | 0.80 | 0.61 | 0.00 | 0.29 | 0.52 | 0.00 | 0.50 | 0.45 | 0.56 | 0.43 | 0.17 | 1.00 | 0.40 |
| Ours-cl-turn3 | 0.00 | 0.62 | 0.43 | 0.80 | 0.65 | 0.18 | 0.33 | 0.57 | 0.50 | 0.64 | 0.53 | 0.65 | 0.68 | 0.27 | 0.93 | 0.49 |
| Ours-cl-turn4 | 0.20 | 0.57 | 0.46 | 1.00 | 0.66 | 0.18 | 0.44 | 0.56 | 0.33 | 0.71 | 0.57 | 0.65 | 0.68 | 0.25 | 0.93 | 0.51 |

The results presented in Table 5 highlight the robustness and adaptability of our SC-MLLM in a continual learning context. After four iterations of updates, our method does not overfit the Test set and demonstrates significant performance improvements across various object categories in the Test B set. Notably, our approach shows superior performance in categories such as *Laptop (LT)*, *Phone (PH)*, and *Refrigerator (RF)*. This consistency underscores our model's ability to mitigate catastrophic forgetting effectively, ensuring sustained high performance through the integration of the Exponential Moving Average and fine-tuning process.

Table 6: The close-loop experiments. "Fast" and "Slow" represent our method's results for the fast system's direct pose prediction and the slow system's corrected prediction based on expert prompts, respectively. "Mean" represents the average manipulation success rate across all tasks.

| Method | Ref | Take USB out | Close fridge | Close box | Toilet seat down | Unplug charger | Mean |
|---|---|---|---|---|---|---|---|
| ManipLLM | CVPR2024 | 0.45 | 0.90 | 0.55 | 0.95 | 0.55 | 0.68 |
| Ours(Fast) | - | 0.55 | 0.70 | 0.40 | 0.90 | 0.60 | 0.63 |
| Ours(Fast+Slow) | - | 0.85 | 0.95 | 0.75 | 1.0 | 1.0 | 0.91 |

## F  DETAILS OF CLOSE-LOOP EXPERIMENTS

In Section 4.2, we validate that SC-MLLM can effectively perform both open-loop and closed-loop control in multiple tasks. In this section, we provide the detailed settings and experiment table for the closed-loop experiments. Specifically, we use the well-known closed-loop benchmark, RLBench (James et al., 2020), as our closed-loop dataset. Five representative tasks with different manipulation trajectories were selected, including "take USB out of computer," "close fridge," "close box," "toilet seat down," and "unplug charger." We collect 1K episodes for the training set of all tasks and conduct 20 tests in the simulator for each task. The evaluation metrics follow previous works (James et al., 2020), assessing task success rate based on predefined success conditions. For the model input, we utilize the front-view image from a third-person perspective, as this view fully captures the manipulated object without occlusion. The key frame filtering method follows the previous work Shridhar et al. (2023). We use 7-DoF to control the Franka Panda robot arm, including not only the position and rotation of the end-effector but also the opening and closing state of the gripper. As shown in Table 6, we compare our proposed method with the previous MLLM-based SOTA method (Li et al., 2023b) under the same experimental setting. Ours (Fast + Slow) achieves an average success rate of 91%, improving accuracy by 23% compared to ManipLLM. The results demonstrate that our slow system can also perform corrections for closed-loop control, showcasing the generalizability of our method.

