# OpenReview forum: "Self-Corrected Multimodal Large Language Model for Robot Manipulation and Reflection"
_ICLR.cc/2025/Conference — ICLR 2025 Conference Withdrawn Submission_

### Official Review · Reviewer_TLvL · 2024-10-21

**Soundness:** 2
**Presentation:** 2
**Contribution:** 2
**Rating:** 5
**Confidence:** 4

**Summary:**

This paper proposes a self-corrected multimodal large language model (SC-MLLM) for manipulation. SC-MLLM contains a fast system for end-effector pose prediction and a slow system for failure reflection and correction. SC-MLLM updates a policy predicting the pose, reflecting the failure, and correcting failure. SC-MLLM trained a policy in the SAPIENCE simulation in experiments. It was validated using RLBench. SC-MLLM outperformed other baselines in many cases of object manipulation. The explanation of the proposed method has a space to improve its clarity.

**Strengths:**

- This paper tackles the issue of hierarchical manipulation learning with motion control (fast system) and task planning (slow system) using LLM, which is significant in the robotics community.
- The experimental results support the usefulness of the proposed method compared to other state-of-the-art manipulation learning methods.
- The effects of expert type, correction times, and continuous policy learning times are investigated in the ablation study. The proposed method is evaluated in the real world.

**Weaknesses:**

- The overview of the SC-MLLM framework, such as how the vector on the right of LLM and one on the left side are used, needs to be clarified.
- The proposed method uses the term "multimodal." However, the used LLM looks like a visual LLM. The proposed SC-MLLM may leverage the interoceptive sense, such as the end effectors' pose. It needs to be clarified why the proposed method contains "multimodal."
- Manipulation success ratios are low in some cases. It needs to be clarified why they are low and how to improve them.

**Questions:**

- How do we get the end-state image? Does the fast system generate it from the initial-state image and language prompts? Or did the authors have the images for the end state in advance?
- What is the left vector of LLM in the left part of Figure 2 and the right vector?
- How to update the Adapter? How can we use it in the inference of the proposed method?

---

### Official Review · Reviewer_NPJu · 2024-11-02

**Soundness:** 2
**Presentation:** 2
**Contribution:** 2
**Rating:** 5
**Confidence:** 3

**Summary:**

The paper proposes a fine-tuned MLLM capable of predicting action deltas through language and reflecting on failed actions to adjust its initial action pose predictions. The paper also demonstrated how this approach can work in both simulation and real-world through experiments, and how it could self-correct for itself.

**Strengths:**

Inspired by a human-like thinking paradigm, the paper fine-tunes its model with simulation data to achieve two goals: rapid end-effector pose prediction and reflective correction of failed actions. The approach is validated through experiments in both simulation and real-world settings.

**Weaknesses:**

1. **Modularity in Expert Feedback and Necessity of Failure Reasoning Fine-Tuning**: If failure reasoning is limited to modular corrections rather than conditioned on current state-action dynamics, fine-tuning for failure reasoning might indeed seem redundant. An instruction-tuned multi-turn MLLM could more effectively handle iterative corrections, especially if trained on curated datasets of failure cases and TAMP-based or zero-shot robotic re-corrections. This approach could lead to more fluid, context-aware adjustments.

2. **Instruction-Tuning with Curated Failure Data**: Training with failure cases and corrective actions, especially from automated systems like Manipulate-Anything, VoxPoser, or ManipGen, could enhance robustness by leveraging real-world re-correction patterns. Using curated multi-turn data could also strengthen the MLLM’s generalization and make its responses more responsive to contextual nuances in multi-step tasks.

3. **Trajectory-Based Data Instead of Keypoint and Motion Primitives**: Generating complete trajectories rather than relying on keypoints or motion primitives could lead to richer data that captures temporal dynamics, allowing the model to learn action-based representations. This may yield more adaptable and generalizable policies that better reflect real-world, continuous task execution.

4. **Necessity of Co-Training for General Knowledge Retention**: If the model is to be considered an MLLM, co-training could be essential for retaining broad knowledge. Demonstrating the model’s retention of general knowledge could be useful, especially if experiments are included that show robust performance across diverse non-specialized tasks.

5. **Focusing Related Work on MLLM and Vision-Language-Action Models**: Narrowing the scope of related work to robotics systems that leverage MLLMs and Vision-Language-Action models would provide a more targeted comparison and avoid diluting the relevance with overly broad robotics literature.

6. **Additional Related Works on Failure Reasoning and Policy Correction**: Incorporating works like AHA, RACER, and Sentinel into related work would provide a more comprehensive view of current methods in failure reasoning and policy correction, offering a direct context for positioning this approach within the landscape of existing research.

**Questions:**

1. **6DoF Pose Representation in Language without Special Tokens**: Representing 6DoF poses solely through language is challenging, as precise spatial and rotational nuances might be difficult to capture effectively without a structured format or specialized tokens. Language can broadly describe an end-effector’s spatial orientation and position, but achieving the specificity required for fine-grained tasks (like robot manipulation) typically benefits from structured representations or direct numerical inputs alongside language.

2. **Comparison with VLA + AHA (https://aha-vlm.github.io/) **: Leveraging an off-the-shelf VLM like VLA for action prediction combined with AHA for failure reasoning might offer a generalized approach with robustness in failure detection, but could lack the specialized finetuning required for accurate 6DoF pose prediction. The custom finetuning approach described may offer more specialized adjustments for pose and corrective actions, while AHA could excel in broader, non-task-specific failure reasoning without extensive re-training.

3. **Generalization and Sim2Real Transfer**: A smaller, simulation-only finetuning dataset raises concerns regarding out-of-domain generalization, particularly for complex 6DoF action prediction tasks where domain discrepancies (like lighting, object variability, and dynamics) between simulation and real-world environments can affect performance. Effective Sim2Real transfer, especially in pose-sensitive tasks, typically requires data that accounts for camera calibration and environment variations to mitigate these gaps. Without real-world data for finetuning, the model might require further adaptation for high-accuracy, real-world applications where visual grounding alone may not suffice.

---

### Official Review · Reviewer_W8TT · 2024-11-03

**Soundness:** 3
**Presentation:** 2
**Contribution:** 3
**Rating:** 5
**Confidence:** 4

**Summary:**

The paper proposes a framework for Self-Corrected Multimodal Large Language Model (SC-MLLM) aimed at improving robotic manipulation by integrating fast and slow reasoning systems. The fast system efficiently predicts end-effector poses for tasks, while the slow system addresses execution failures by identifying error causes and seeking expert feedback for corrections. In summary, SC-MLLM enhances robotic manipulation across different simulator with closed loop feedback and in real-world experiments.

**Strengths:**

1. Integration of slow and fast systems using LLMs: The fast system predicts pose and slow system predicts failure correction from user feedback.

1. Sim-to-real gap: The experiments in both simulated environments (SAPIEN, RLBench) and real-world settings demonstrate applicability of the proposed system.

1. Continuous Learning: The model adapts based on corrected samples to improve performance over time and reduce reliance on expert interventions.

1. Real-World Applicability: the paper discusses sim-to-real gap and the implications of design choices.

**Weaknesses:**

1. reliance on expert knowledge for position and rotation: the design of slow system requires expert knowledge for exact position and rotation of the robot - it is unclear how feasible it is to obtain in the real world.
1. limited reasoning in slow system: for long horizon, open-ended tasks with multiple optimal/feasible solutions.
1. specific form of failure recovery: slow system performs recovery in specific set of scenarios where the reach is incorrect in a substep. Does this cover all possible failure scenarios? there is lack of discussion on failure analysis.

**Questions:**

1. Why should the expert know the required position and rotation coordinates? Are there any information gathering actions involved? How does the slow system “search” over possible solutions to propose any answer?

1. End-state based decision - does not account for motions where the manner of trajectory matters? OR long horizon task with subgoals?

1. “For position errors, we assess whether the contact point falls within the object’s affordance region. For rotation errors, we evaluate whether the angle between the predicted Z axis and the normal of the object’s movable plane exceeds 30 degrees.” - Indicate the implications and limitations of this approach.

---

### Note · Authors · 2024-11-14

I have read and agree with the venue's withdrawal policy on behalf of myself and my co-authors.